# Apolipoprotein-L Functions in Membrane Remodeling

**DOI:** 10.3390/cells13242115

**Published:** 2024-12-20

**Authors:** Etienne Pays

**Affiliations:** Laboratory of Molecular Parasitology, Institut de Biologie et de Médecine Moléculaires (IBMM), Université Libre de Bruxelles, 6041 Gosselies, Belgium; etienne.pays@gmail.com

**Keywords:** APOL1 nephropathy, APOL1 risk variants, APOL3 antibacterial activity, adipogenesis, angiogenesis, antigen cross-presentation, kidney disease, membrane fission, membrane fusion, mitophagy

## Abstract

The mammalian Apolipoprotein-L families (APOLs) contain several isoforms of membrane-interacting proteins, some of which are involved in the control of membrane dynamics (traffic, fission and fusion). Specifically, human APOL1 and APOL3 appear to control membrane remodeling linked to pathogen infection. Through its association with Non-Muscular Myosin-2A (NM2A), APOL1 controls Golgi-derived trafficking of vesicles carrying the lipid scramblase Autophagy-9A (ATG9A). These vesicles deliver APOL3 together with phosphatidylinositol-4-kinase-B (PI4KB) and activated Stimulator of Interferon Genes (STING) to mitochondrion–endoplasmic reticulum (ER) contact sites (MERCSs) for the induction and completion of mitophagy and apoptosis. Through direct interactions with PI4KB and PI4KB activity controllers (Neuronal Calcium Sensor-1, or NCS1, Calneuron-1, or CALN1, and ADP-Ribosylation Factor-1, or ARF1), APOL3 controls PI(4)P synthesis. PI(4)P is required for different processes linked to infection-induced inflammation: (i) STING activation at the Golgi and subsequent lysosomal degradation for inflammation termination; (ii) mitochondrion fission at MERCSs for induction of mitophagy and apoptosis; and (iii) phagolysosome formation for antigen processing. In addition, APOL3 governs mitophagosome fusion with endolysosomes for mitophagy completion, and the APOL3-like murine APOL7C is involved in phagosome permeabilization linked to antigen cross-presentation in dendritic cells. Similarly, APOL3 can induce the fusion of intracellular bacterial membranes, and a role in membrane fusion can also be proposed for endothelial APOLd1 and adipocyte mAPOL6, which promote angiogenesis and adipogenesis, respectively, under inflammatory conditions. Thus, different APOL isoforms play distinct roles in membrane remodeling associated with inflammation.

## 1. APOL Families

Apolipoproteins-L (APOLs) are mammalian lipid-interacting proteins encoded by rapidly evolving multigene families, and present at various levels in all organs (https://www.proteinatlas.org/search (accessed on 17 December 2024)). Phylogenetic trees of APOLs reveal frequent duplications and diversification, illustrated by important differences in isoform numbers between species [1,2,3]. Whereas the human family contains six members, the murine family contains 12. The first identified APOL member is APOL1, which was found to be associated in the blood with the densest fraction of High-Density Lipoprotein particles (HDL-C) that are involved in cholesterol recycling [4]. Therefore, APOL1 function was considered as involved in lipid transport. The different APOL isoforms share a similar structure, with several α-helices that include putative transmembrane spans. Only the structure of the human APOL1 and APOL2 N-terminal domain was determined, and it is characterized by a non-classical four-helix bundle motif [5]. Among the human and murine members, significant information is available for APOL1, APOL3, APOL7C and mAPOL6 [3,6]. As discussed in this review, this information points to functions linked to the control of membrane dynamics, particularly under inflammatory conditions.

## 2. Distinctive Features of Human APOL1 and APOL3

### 2.1. APOL1-Specific Folding and Transmembrane Insertion Linked to Extracellular Activity

As shown in Figure 1, APOL1 and APOL3 exhibit a similar arrangement of α-helices on both sides of a putative transmembrane (TM) hairpin helix.

Only the structure of the N-terminal APOL1 domain is known [5]. This domain is organized into five helices, with helices 2 to 5 being folded in a four-helix bundle. To account for their prominent characteristics, helices 2 and 4 are termed here as Hydrophobic Cluster-1 and Leucine Zipper-1 (HC1 and LZ1, respectively). Similar helices are present in APOL3, suggesting a similar folding. Helix 5 is probably involved in interactions with anionic phospholipids, because it contains stretches of positively charged residues typically responsible for such interactions. In the isolated N-terminal domain, helix 5 can adopt two configurations [5], but in the entire protein, helix 5 anchoring to the TM hairpin probably fixes it in the open configuration, which allows both helix 4 (LZ1) interaction with LZ2 and helix 5 interaction with other proteins (see Section 2.2).

Immediately downstream from helix 5, two hydrophobic helices folding in a hairpin-like structure are potentially able to cross a membrane, forming a TM pore. In the case of APOL1 only, TM insertion absolutely requires acidic conditions, due to the APOL1-specific requirement of protonation of acidic residues to allow hydrophobic interactions [7]. Once inserted, APOL1 exhibits weak anion conductance at acidic pH, but high cation conductance at neutral pH [8,9,10]. Thus, APOL1 pore-forming activity is pH-gated, whereas APOL3 can form cation pores irrespective of the pH.

Next to the TM pore, a Membrane-Addressing Domain (MAD) [10] exhibits putative phospholipid-binding stretches like helix 5, and can fold in a two-stranded helix with a potential Cholesterol Recognition Amino Acid Consensus (CRAC-1) [11,12]. At the C-terminus, helices able to fold into a hairpin contain a highly hydrophobic stretch (HC2) harboring another CRAC motif (CRAC-2), as well as residues conferring pH-gating to the ionic pore [13]. Whether such pore-controlling activity implies transmembrane HC2 insertion is debated [14]. The C-terminal hairpin also contains a leucine zipper helix (LZ2). In APOL1 only, LZ-driven cis interaction occurs between N- and C-terminal helices [15], preventing HC1 and CRAC-2 exposure. This APOL1-specific interaction is disrupted under acidic conditions [16], allowing both HC1 and CRAC-2 exposure at low pH only.

Interestingly, APOL2, which is the closest homologue of APOL1 but lacks a signal peptide (61.4% identity/74.6% similarity), exhibits significant differences from APOL1 in both CRAC-2 and LZ2, implying that in contrast to APOL1, APOL2 is devoid of CRAC-2, and is unlikely to fold through LZ2-LZ1 interaction (Figure 2). Accordingly, in yeast two-hybrid interaction experiments, the APOL2 N-terminal domain, with APOL1-like LZ1 and helix 5 (respectively 85 and 93% similarity), only poorly interacted with APOL1 LZ2 (6% compared to APOL1 cis interaction) [15]. In contrast, the APOL1 N-terminal domain strongly interacted with APOL2 LZ2 (90% compared to APOL1 cis interaction) [15], suggesting that in APOL2, like in APOL3 [16], helix 5 interacts with LZ2 (Figure 2).

As far as we know, APOL1 is the only APOL member exerting an extracellular function. Besides the intracellular isoforms generated by the differential splicing of APOL1 transcripts, an APOL1 isoform is provided with an N-terminal signal peptide that allows secretion. This secreted isoform kills the bloodstream African parasite *Trypanosoma brucei*, which is responsible for lethal infection in different mammals, such as the Nagana disease in cattle [17]. Secreted APOL1 circulates in the bloodstream through its association with HDL-C particles, which are involved in cholesterol recycling. Such association may result from the cholesterol-binding activity of CRAC-1, because CRAC motifs appear to be absent in all other known APOLs, which are only intracellular [12]. Circulating APOL1 is also contained in lipid-poor multiprotein complexes, which, like HDL-C particles, contain Haptoglobin-Related Protein (HPR) [18]. HPR is the key for APOL1 uptake in trypanosomes, through the specific recognition of HPR–hemoglobin complexes by a trypanosome surface receptor [19]. Interestingly, the HPR N-terminal sequence, which drives HPR association with HDLs [20], contains a CRAC motif (15-RQLFALYSGNDV-26). Thus, APOL1 and HPR may circulate in the blood through their common association with cholesterol in lipoprotein complexes. As HPR–hemoglobin complexes are efficiently taken up in trypanosomes as growth factors [19], HPR-associated APOL1 is avidly taken up by the parasite. Once internalized in the parasite acidic endosomes, APOL1 inserts into endosomal membranes, generating a weak transmembrane flux of anions [10]. This pore-forming activity does not directly cause trypanosome lysis, but transmembrane insertion indirectly induces the apoptotic-like death of the parasite [21]. Indeed, trypanosome death results from mitochondrial outer membrane permeabilization, which is linked to the intracellular traffic of APOL1-containing endosomal membranes, notably to the mitochondrion. Interestingly, this traffic is conducted by the TbKIFC1 kinesin, which is responsible for cholesterol clearance from the parasite surface [21,22]. As the CRAC-2 motif is predicted to become accessible under acidic conditions, CRAC-2 may participate in TbKIFC1-mediated transport of APOL1-carrying endosomes.

In conclusion, the two original features of APOL1, pH-dependent cis interaction and pH-dependent transmembrane insertion, are linked to the exclusive extracellular activity of the secreted isoform, after its uptake in the acidic medium of trypanosome endosomes.

### 2.2. Impact of APOL1 Folding on Interactions with Other Proteins

Only two proteins were identified as APOL1 binders (Figure 3). In immunoprecipitation experiments, intracellular APOL1 was stoichiometrically associated with each of the three subunits of Non-muscular Myosin-2A (NM2A) [15]. Accordingly, APOL1 can bind in vitro to the NM2A Regulatory Light Chain (RLC) (E. Pays, unpublished). In an experimental interaction assay system in *Escherichia coli*, APOL1 was also found to interact with the mitophagy receptor Prohibitin-2 (PHB2) [16], mimicking murine APOL9 interaction with the same protein [2]. This interaction may involve helix 5 [16]. APOL3 does not bind to PHB2, probably because in APOL3, helix 5 interacts with LZ2 [16].

Once extracellular APOL1 is taken up in *T. b. rhodesiense* endosomes, its C-terminal domain (HC2-LZ2) interacts with the *T. b. rhodesiense*-specific protein SRA (Serum Resistance-Associated protein) [23]. This interaction prevents APOL1 insertion into endosomal membranes, allowing *T. b. rhodesiense* to escape from APOL1 lytic activity and consecutive parasite infectivity in humans, thereby causing sleeping sickness in east Africa [24].

The APOL1 interaction profile is clearly influenced by its specific folding, because disruption of LZ2-LZ1 cis interaction, following either LZ2 deletion or LZ2 mutations, allows strong binding of PI4KB to intracellular APOL1 and increases APOL3 interaction with APOL1, inhibiting APOL3 activity [15,16]. Natural LZ2 mutations, defined as G1 and G2, are widespread in individuals of west African origin, in whom they correlate with chronic kidney disease susceptibility [25]. These mutations were probably selected because they allow APOL1 to escape neutralization by *T. rhodesiense* SRA, but they also disrupt LZ2-LZ1 cis interaction, triggering enhanced exposure of HC1 and HC2 and, hence, increased APOL1 hydrophobicity [15]. This higher hydrophobicity causes podocyte dysfunctions resulting from PI4KB and APOL3 inactivation at the Golgi, characterized by pedicel effacement, change of cellular architecture and increased mobility linked to actomyosin reorganization (kidney disease hit 1) [15]. In addition, the G1- or G2-specific exposure of HC2 allows increased accessibility of HC2-contained CRAC-2, explaining the enhancement by G1 or G2 of cholesterol clustering in both HDL-C particles and tissues [26,27]. CRAC-2 exposure could also account for the toxic surface cation fluxes linked to the inflammatory stage of APOL1 nephropathy (kidney disease hit 2), probably through activation of podocyte cholesterol-sensitive cation channels [12] (see Section 4).

### 2.3. APOL3 Interactions

APOL3 does not appear to be folded like APOL1, presumably due to a lower interaction potential between LZ1 and LZ2 [15,16]. Instead, APOL3 LZ2 can interact with helix 5 [16]. Probably because the APOL3 N-terminal domain is more accessible than that of APOL1, this domain exhibits specific interactions with different proteins. These proteins are all involved in the control of PI(4)P synthesis, and include the PI4KB kinase together with the PI4KB-controlling factors NCS1, CALN1 and ARF1 [15,16] (Figure 4). The three-helix 2–4 bundle of APOL3 interacts with a similar helix bundle in the regulatory domain of PI4KB [16]. APOL3 and PI4KB form a trimer complex with either NCS1 or CALN1, depending on the calcium concentration, and ARF1 could interfere with this trimer following its activation (conversion of bound GDP into GTP).

In addition, APOL3 can also interact with the endosomal Vesicle-Associated Membrane Protein-8 (VAMP8) [16]. This interaction involves the helices flanking the transmembrane hairpin helix (helices 4–5 and MAD) [16] (Figure 4). Probably because helix 5 interacts with LZ2 when not interacting with VAMP8, no protein binding to APOL3 LZ2 has been identified so far [15,16].

## 3. APOL1 and APOL3 Basic Functions

### 3.1. Intracellular APOL1: Inflammation-Linked Vesicular Trafficking

The experimental deletion of APOL1 in cultured podocytes (APOL1 KO) does not induce a significant phenotype, in keeping with the apparent absence of pathology in human individuals naturally lacking APOL1 [15,28,29]. However, under inflammatory signaling induced by poly(I:C), a viral mimetic, the absence of APOL1 was associated with a strong reduction in mitophagy and apoptosis [15,16]. Normally, the viral or poly(I:C)-mediated induction of the type-I interferon (IFN-I) pathway promotes the traffic of vesicles carrying the lipid scramblase ATG9A from the Golgi to MERCSs for the induction of autophagy, mitophagy and apoptosis [30,31,32,33,34,35]. APOL1 is associated with ATG9A vesicles together with APOL3 and PI4KB, and appears to be required for this traffic [16]. Thus, ATG9A traffic inhibition may explain the inhibitory effect of APOL1 KO on mitophagy and apoptosis. Interestingly, such traffic is also triggered upon APOL3 deletion (APOL3 KO) [16], and APOL1 also appears to be required in this case, because APOL1 deletion in APOL3 KO podocytes (APOL3 + APOL1 double KO) fully prevented PI4KB delocalization from the Golgi [15]. The APOL1 function in ATG9A traffic is consistent with its association with NM2A and PHB2, because NM2A is known to drive ATG9A vesicle traffic [36,37,38] and PHB2 directs cargoes trafficking to MERCSs for mitophagy initiation [39].

In conclusion, APOL1 does not directly participate in mitophagy and apoptosis, but it is indirectly involved in these processes through its ability to control ATG9A vesicle trafficking. As detailed in Section 3.3, the lack of APOL1 activity in mitophagy and apoptosis results from the APOL1-specific inability to perform transmembrane insertion at neutral pH.

### 3.2. APOL3: Indirect Role in Membrane Fission

Unlike APOL1 deletion, APOL3 absence, whether in cultured podocytes or in individuals, induces a significant phenotype even under non-inflammatory conditions [15,16,40,41]. In cultured podocytes, this complex phenotype involves the dissociation of PI4KB from the Golgi, PI4KB trafficking in ATG9A vesicles and the induction of mitophagy and apoptosis, but in an abortive process [15,16]. Thus, APOL3 KO mimics the IFN-I-mediated activation of ATG9A vesicle traffic, but does not allow completion of mitophagy and apoptosis, indicating that APOL3 must also play a role in these processes. These observations suggest two steps in APOL3 activity.

(1)APOL3 prevents PI4KB dissociation from the Golgi, and either IFN-I-signaling or APOL3 KO disrupts this control, allowing PI4KB traffic in ATG9A vesicles. IFN-I signaling promotes ARF1 activation through the conversion of ARF1-bound GDP to GTP [30], which increases ARF1 binding to PI4KB [42]. Because ARF1 can interact with the same PI4KB three-helix bundle as that binding to APOL3, ARF1 and APOL3 interactions with PI4KB may be mutually exclusive [16]. Therefore, ARF1 activation may disrupt the PI4KB-APOL3 complex, achieving a similar effect as APOL3 deletion or inactivation.(2)APOL3 KO-linked inability to complete mitophagy and apoptosis could result from the absence of ARF1 activation. Indeed, ARF1 activation is required for mitochondrial membrane fission [32], which initiates mitophagy and apoptosis [43,44].

APOL3 binds to a three-helix bundle of the PI4KB regulatory domain, and also interacts with the PI4KB activity controllers NCS1, CALN1 and ARF1 [16]. Whereas NCS1 allows PI4KB activation in the presence of calcium, CALN1 inhibits PI4KB in the absence of calcium, and activated ARF1 can stimulate PI4KB [6,16,30,42]. In vitro, APOL3 can directly activate PI4KB in the presence of calcium, independently of the presence of NCS1, and, in contrast, NCS1 does not activate PI4KB, even with calcium [15,16]. Thus, in the intracellular context, NCS1 might not activate PI4KB directly, but indirectly through local calcium delivery to PI4KB at the Golgi membrane. Conversely, CALN1 association with PI4KB prevents PI4KB activation under low-calcium conditions. Finally, activated ARF1 could disrupt the APOL3-PI4KB-NCS1 complex, triggering PI4KB dissociation from the Golgi.

At the Golgi, PI4KB activity promotes membrane recruitment of the fission factor NM2A and associated membrane-bending factors, and is linked to actin polymerization at the fission site [45,46]. PI(4)P, possibly clustered in lipid raft-like microdomains, recruits Golgi phosphoprotein-3 (GOLPH3), a PHB2-associated trans-Golgi and mitochondrial protein that binds to both F-actin and the NM2A binder myosin-18A (MYO18), coupling fission factors with actomyosin for the generation of a pulling force driving membrane fission [47,48,49]. Thus, PI4KB is involved in the initiation of vesicle budding and exit from the Golgi, which is linked not only to secretion, but also to cancer progression, notably through the promotion of metastasis [3,50,51].

At MERCSs, PI4KB is involved in mitochondrial membrane fission, which is linked to the induction of mitophagy, autophagy and apoptosis [31,32,33,34,35,52,53]. At the mitochondrial membrane, lipid raft-like microdomains recruit fission factors such as Fission factor-1 (FIS1) and Dynamin-Related Protein-1 (DRP1), together with the fusion factor Syntaxin-17 (STX17) [32,54,55,56,57]. Because mitophagy depends on membrane fission independently of DRP1 [43,58], the role of PI4KB in mitophagy could be the organization of PI(4)P-enriched microdomains allowing recruitment of FIS1, which is involved in stress-induced fission and mitophagy [59]. FIS1 could oligomerize at lipid rafts through its ability to bind anionic phospholipids [60]. This hypothesis is in keeping with the observation that PI4KB co-localizes with FIS1, in a process increased following APOL3 loss or inactivation [16].

In natural APOL3 KO individuals, the function of kidney podocytes is severely altered, and APOL3 KO can aggravate podocyte dysfunctions due to expression of the APOL1 risk alleles G1 or G2 [40,41]. Thus, APOL3 KO mimics the G1 or G2 effects, but with increased intensity. This phenotype is consistent with the observation that G1 and G2 inactivate APOL3 [15], but less severely than APOL3 KO.

In conclusion, the synthesis of PI(4)P by PI4KB is crucial for membrane fission at both the Golgi and MERCSs, and this activity involves distinct controls by APOL3 and APOL1. Accordingly, interference with APOL3 activity, such as that which occurs following expression of the APOL1 variants G1 or G2, includes podocyte dysfunctions resulting from PI4KB activity deregulation, notably through actomyosin reorganization and increased cellular motility [3,15].

### 3.3. APOL3: Direct Role in Membrane Fusion

APOL3 deletion or inactivation does not only inhibit mitochondrion fission, but also mitophagosome fusion with endolysosomes [16]. This observation can be readily explained by the capacity of APOL3 to promote vesicular fusion. APOL3 can interact with the fusogenic SNARE (Soluble-N-ethylmaleimide-sensitive Attachment Receptor) protein VAMP8, which is present at the endosomal membrane [16]. VAMP8 is known to interact with the SNARE proteins STX17 and SNAP29 (Synaptosomal-Associated Protein-29) to promote the fusion between endolysosomes and autophagosomes [61]. However, the SNARE partner(s) of VAMP8 for endolysosome fusion with mitophagosomes remained unknown [62]. APOL3 appears to fulfill this function, because APOL3 association with vesicles containing mitochondrion-specific lipids promoted the fusion of these vesicles with VAMP8-carrying vesicles containing endosomal-specific lipids [16]. APOL3’s ability to insert into membranes at neutral pH must be involved in the fusion process, because APOL1, which only exhibits transmembrane insertion at low pH, required a pretreatment at acidic pH to induce fusion with VAMP8 vesicles [16]. Furthermore, in trypanosomes, APOL1 promotes the fusion of mitochondrial membranes after traffic in acidic endosomes [21]. As a trypanosome homologue of VAMP8 (TbVAMP7B) is involved in APOL1 trypanolytic activity [63], TbVAMP7B may participate in APOL1-carrying endosome–mitochondrion fusion, mirroring podocyte VAMP8-APOL3 interaction.

That APOL3 may exert a fusion activity analogous to that of STX17 is supported by functional similarities between these proteins, despite their high sequence difference. In both STX17 and APOL3, the transmembrane hairpin helices are highly flexible, owing to their high proportion of β-branched (Ile/Val) and helix-breaking (Gly/Pro/Ser/Thr) amino acids. Hydrophobic helix flexibility is crucial for transmembrane helices to drive membrane fusion [64]. Moreover, like APOL3, STX17 exhibits Golgi-to-mitochondrion trafficking and activity in both mitochondrial fission and auto-/mitophagy initiation [55,56,65,66]. However, STX17 is not directly involved in the fusion between mitophagosomes and endosomes [62], and its exact function in mitophagy is still unclear. The dissociation of STX17 from interaction with FIS1 triggers aberrant STX17 accumulation and oligomerization on mitochondria, which can induce mitophagy [56]. Whether this process is relevant to APOL3-mediated mitophagosome fusion is unknown.

APOL3 may be specifically targeted to mitophagosomes due to its strong binding to cardiolipin [15], which is incorporated into the mitophagosome membrane [67,68,69,70]. Significantly, VAMP8 also binds to cardiolipin, and cardiolipin was strongly involved in promoting vesicle fusion between APOL3- and VAMP8-carrying vesicles [16].

In conclusion, through their capacity to interact with both cardiolipin and VAMP8, APOL3 and APOL1 can promote fusion between mitochondrial and endosomal membranes, although APOL1 requires acidic conditions for such activity.

### 3.4. Respective Roles of APOL1 and APOL3 in Apoptosis

APOL1 and APOL3 activities in mitochondrial dynamics may also account for their apoptotic activity [7,15,16,21,44]. According to their respective activities, APOL1 could affect podocyte apoptosis through its involvement in APOL3 traffic to the mitochondrion, whereas APOL3 transmembrane insertion could be involved in megapore formation through its combined activities in mitochondrial membrane fusion and cation-pore-forming activity. K^+^ efflux is known to activate apoptosis [71], but it is not known whether APOL3-driven transmembrane K^+^ conductance, as occurs in vitro [7], is involved in APOL3-mediated apoptosis.

In trypanosomes, recombinant APOL1 or APOL3 can induce an apoptotic-like process without evidence for increased membrane fission, and APOL-induced mitochondrion fusion does not seem to be linked to parasite death [7,21]. Therefore, trypanosome apoptosis could occur through endogenous apoptotic-like activity induced following APOL transmembrane insertion, possibly due to cation-pore-forming activity.

## 4. APOL1 and APOL3 Roles in Kidney Disease

The key involvement of the APOL1 risk variants G1 and G2 in non-diabetic kidney disease has been amply detailed and commented on in a recent review [12]. In addition to the effects of these variants on podocyte biology under non-inflammatory conditions, particularly regarding actomyosin control and mitochondrial function, as detailed in Section 2 and Section 3, inflammation was found to induce major podocyte cytotoxicity. When studied in various experimental gene expression assays in cultured cells, including human podocytes, the specific G1- or G2-linked toxicity was consistently linked to surface cation fluxes induced by extracellular (secreted) APOL1. Given the ability of APOL1 to form cation pores in trypanosomal or synthetic membranes, many investigators concluded that as compared to WT APOL1, the secreted G1 and G2 variants can better insert into the podocyte plasma membrane, causing increased fluxes of both K^+^ and Ca^2+^, thereby inducing stress-driven cytotoxicity.

Contradicting the pore-forming hypothesis, all studies analyzing APOL1 cation conductance concluded that this process strictly requires acidic conditions, as detailed in Section 2.1. Such conditions are not present at the cell surface, rendering APOL1 transmembrane insertion very improbable. An alternative view is that given their increased hydrophobicity, including enhanced CRAC-2 exposure (see Section 2.1), G1 and G2 exhibit increased interaction with cholesterol at the podocyte plasma membrane, stimulating cation fluxes by the cholesterol-sensitive TRPC6 and BK channels, which conduct Ca^2+^ and K^+^, respectively, and play key roles in podocyte filtration activity [12]. Such G1 or G2 interaction with cholesterol at the podocyte surface requires HDL-C or at least cholesterol level reduction in the extracellular environment, because secreted APOL1 is normally tightly sequestered by the serum cholesterol-rich lipoprotein particles. This condition is met in the case of inflammation, in which HDL-C levels are lowered concomitantly with an increase in APOL1 expression, and it also occurs when cultivating cells in vitro, when HDL-C levels are low and APOL1 levels are often higher than normal. Interestingly, a natural mutation (N264K) appears to ablate G1 or G2 toxicity. Since this mutation is predicted to inactivate the CRAC-2 motif [12], it is tempting to propose that CRAC-2 interaction with cholesterol is crucially involved in podocyte cytotoxicity.

So far, no direct evidence for APOL1 insertion in the plasma membrane has been provided, and the role of G1 or G2 in activating TRPC6 and BK channels also remains to be demonstrated.

This issue has some clinical implications. Whereas neutralizing surface-exposed APOL1 with specific APOL1-targeting proteins or drugs appears to reduce the disease [12], totally inactivating APOL1 is expected to increase the patient’s sensitivity to pathogen infection. Indeed, intracellular APOL1 is required for mitophagy induction to limit inflammation-driven mitochondrial damage. Thus, the treatment of kidney disease should be limited only to the neutralization of G1 or G2 present at the podocyte surface.

## 5. APOL1 and APOL3 Roles in Pathogen Infection

### 5.1. STING Activation and Degradation

IFN-I inflammatory signaling induced by viral, bacterial or parasitic infection is initiated by activation of the pathogen DNA-sensor STING (Stimulator of Interferon Genes), a transmembrane ER protein. Whereas STING activation depends on its translocation from the ER to the Golgi for oligomerization at the Golgi surface, STING degradation and, hence, inflammation termination, depends on STING translocation from the Golgi to endolysosomes, probably via transit through MERCSs [72] (Figure 5).

STING translocation from the ER to the Golgi results from the transient lowering of free cholesterol levels due to cholesterol esterification induced by cyclic GMP-AMP (cGAMP), a messenger produced by cyclic GMP-AMP synthase (cGAS) in response to the detection of viral or bacterial DNA in the cytosol [73]. Infection-linked transient decline in ER cholesterol levels disrupts STING–cholesterol interactions [73] and promotes ER membrane fluidity and dynamics, allowing STING binding to the PI4KB product PI(4)P, which is required for STING activation [74,75,76]. PI(4)P induces membrane curvature and allows selective STING targeting to Golgi membranes, where PI4KB activity is concentrated.

Once in the Golgi, STING undergoes oligomerization, possibly on PI(4)P-rich microdomains. This oligomerization triggers the induction of gene transcription for IFN-I signaling, through phosphorylation-induced recruitment and phosphorylation of the transcription factor Interferon Regulatory Factor-3 (IRF3) [77]. Given the tight interactions of both PI4KB and PI(4)P with APOL3 and the crucial role of APOL3 in PI4KB activity control at the Golgi [15,16], APOL3 must be involved in STING activation.

Like STING, APOL1 can also interact with both cholesterol and PI(4)P. Thus, the transient reduction in ER cholesterol could allow not only STING, but also APOL1 traffic from the ER to the Golgi. Accordingly, poly(I:C) treatment increases the Golgi/ER ratio of APOL1, contrasting with a Golgi/ER APOL3 ratio decrease under the same conditions [15].

Following the STING-mediated activation of immune transcription factors, inflammatory IFN-I signaling activates ARF1, which triggers PI4KB, APOL1 and APOL3 dissociation from the Golgi and trafficking in Golgi-derived ATG9A vesicles to mitochondrial and endosomal membranes, for the induction and completion of auto/mitophagy and apoptosis [16,30,31]. Strikingly, this is concomitant with STING translocation in Golgi-derived ATG9A vesicles to MERCSs and endolysosomes, which is also linked to autophagy induction and inflammation termination [78,79,80,81,82,83,84,85,86,87].

Whereas PI4KB and ARF1 are crucially involved in membrane fission for auto/mitophagy initiation [31,32], APOL3 and STING are both involved in auto/mitophagosome fusion with endolysosomes for auto/mitophagy completion [16,87]. Golgi-derived ATG9A vesicle traffic appears to be controlled by APOL3, because it can be triggered in the absence of inflammatory signaling when APOL3 is lost or inactivated [15,16]. Thus, APOL3 could control not only PI4KB, but also STING sequestration at the Golgi.

In addition to sharing common trafficking pathways, APOL3 and STING also exhibit related characteristics. Both proteins are involved in membrane fusion. Specifically, STING interacts with the autophagic SNARE fusion protein STX17 [87], whose function appears to be played by APOL3 in the case of mitophagy [16]. Both STX17 and APOL3 contain SNARE-like helices that can interact with phospholipids, but whether STING and APOL3 can interact is not known.

In conclusion, APOL1 and APOL3 appear to be involved in ER-to-Golgi membrane-trafficking linked to STING activation for induction of inflammation, and in Golgi-derived STING trafficking linked to the initiation and completion of autophagy for inflammation termination. In addition, inflammation-linked K^+^ efflux induced by APOL1 variants in the plasma membrane [12] may trigger inflammasome activation and consecutive stress signaling [88]. These considerations provide an explanation for the key involvement of STING and inflammasome in APOL1 nephropathy, even without inflammation induced by infection [89].

### 5.2. Antigen Cross-Presentation in Dendritic Cells

Cross-presentation of phagocytosed antigens at the surfaces of dendritic cells (DCs) is required to trigger T cell responses against many pathogens and tumors. This process is promoted through the prevention of phagosome fusion with endolysosomes, and involves antigen release from the phagosome to the cytoplasm before endocytic digestion [90]. In mice, antigen cross-presentation requires APOL7C, a murine APOL isoform strongly induced by IFN-I-mediated inflammation in DCs [91,92]. Interestingly, human APOL3, which shares 40.2% identity and 55.2% similarity with APOL7c (Figure 6), was similarly able to initiate phagosomal permeabilization [92].

Murine APOL7C is unlikely to be the human APOL3 homologue, because murine APOL8 exhibits a higher homology with APOL3 (41.6% identity; 59.6% similarity), and appears to share APOL3’s ability to interact with NCS1 [6]. Moreover, unlike APOL3, APOL7C is characterized by the insertion of an acidic stretch clearly separating helices 2–3 from helix 4 (Figure 6). Consequently, the three-helix 2–4 bundle, which can interact with both NCS1 and PI4KB [16], is probably disrupted, likely hindering these interactions.

Despite this difference, APOL7C could be induced by IFN-I signaling to delocalize with PI4KB and activated ARF1 in Golgi-derived vesicles to phagosomes, like what appears to occur in macrophages [93]. Therefore, it is tempting to speculate that following IFN-I-triggered traffic from the Golgi to phagosomes, PI4KB and activated ARF1 would induce PI(4)P synthesis for phagosomal membrane fission. Like what occurs with mitophagosomes in mitophagy, phagosome membrane fission is probably coupled to fusion with endolysosomes for the generation of phagolysosomes, because this process is known to require PI(4)P synthesis [94]. Incomplete fission and/or fusion due to the presence of the APOL7C-specific acidic cluster is expected to cause phagosome membrane permeabilization. Given the involvement of helix 4 in APOL3 interaction with endosomal VAMP8 for membrane fusion [16], helix 4 repositioning due to acidic stretch insertion may affect APOL7C fusion with endolysosomes, compromising membrane integrity. The release of antigens during this process may be compared with the release of cytochrome C from mitochondria during incomplete fission/fusion events associated with apoptosis [44].

Upon antigen presentation, DCs are programmed for apoptosis, presumably to limit inflammation [95,96,97]. In mice, the induction of several APOLs, including APOL7C, specifically characterizes CD8α^+^ DCs [91], which perform antigen cross-presentation and exhibit a shorter lifespan than other DCs [98]. Accordingly, APOL7C is involved not only in antigen cross-presentation [92], but also in DC apoptosis induced by poly(I:C) [91]. Therefore, endolysosomal membrane permeabilization linked to incomplete fusion with phagosomes appears to induce both antigen cross-presentation and apoptosis.

In humans, none of the six APOLs exhibits acidic sequence insertion. According to the scenario proposed in mice, antigen cross-presentation in human DCs should involve an APOL isoform inducible by IFN-I, but which should be deficient in fission/fusion activity. The only candidate is intracellular APOL1 (49.2% similarity with APOL7C), which is induced by IFN-I, lacks membrane fusion-promoting activity and is unable to bind NCS1, even at low pH [15,16,91].

In conclusion, antigen release from DC phagosomes and consecutive antigen cross-presentation may require incomplete membrane fission/fusion activity by APOL7C, possibly linked to defective phagolysosome biogenesis. In human cells, APOL1 may perform APOL7C-like activity.

### 5.3. Control of Mitochondrial Damage

Mitophagy and apoptosis are alternative responses to the mitochondrial damage induced by the inflammatory response to infection. Through the removal of dysfunctional mitochondria by mitophagy, cells manage to cope with mitochondrial stress until the damage becomes too great, which leads to the activation of apoptosis [99]. As APOL1 and APOL3 are involved in both mitophagy and apoptosis, interference with APOL1 and/or APOL3 activity results in pathology linked to mitochondrial dysfunction.

Upon APOL3 absence or APOL3 inactivation by APOL1 C-terminal variants, cultured podocytes exhibit incomplete mitophagy linked to the generation of mitochondrial ROS independently of the induction of IFN-I signaling [16]. Moreover, in these cells, poly(I:C)-induced apoptosis nearly disappears, allowing the persistence of dysfunctional mitochondria and abortive mitophagosomes [16,100]. Accordingly, APOL1 G1/G2 expression triggers mitochondrial dysfunction [101,102,103,104], whereas in renal carcinoma cells that are characterized by significant inflammation [105], APOL1 KO triggers severe autophagy and mitochondrion dysfunction [106]. Consistently, in endothelial cells, APOL1 G1/G2 expression induces autophagic dysfunction and mitochondrial stress [107], causing exacerbated sepsis [108]. Because kidney function is particularly affected by interference with mitochondrial fission [109], altered mitochondrial membrane dynamics could participate in APOL1 nephropathy linked to viral infection (HIVAN and COVAN: HIV- and COVID-19-associated nephropathy, respectively) [110,111].

In conclusion, through their key involvement in mitophagy and apoptosis, APOL1 and APOL3 contribute to avoiding the detrimental effects of infection-induced inflammation on mitochondrial function. Interference with these activities causes severe impairment of energy metabolism and triggers auto/mitophagy dysfunctions, notably involved in APOL1 nephropathy linked to viral infection.

### 5.4. Resistance to Intracellular Bacteria

In a mutagenesis screen of 19,050 human genes, APOL3 was selectively identified as a potent bactericidal agent [112]. Through its detergent-like activity on membranes of intracellular bacteria, APOL3 induced the formation of vesicular-like structures with bacterial membrane debris. Thus, the membrane fusion potential of APOL3 is exploited by nonimmune cells to achieve sterilizing immunity. Given the similarity in lipid composition between bacterial and mitochondrial membranes, particularly their high cardiolipin content, it is tempting to propose that the membrane fusion process induced by APOL3 on bacteria involves interaction with endosomal VAMP8, like that which occurs during mitophagy, resulting in the digestion of bacterial residual components in endolysosomes.

In conclusion, the bactericidal activity of APOL3 could involve membrane fusion activity like that exhibited for the completion of mitophagy.

### 5.5. Control of Viral Infection

IFN-I signaling strongly increases APOL1 and APOL3 expression [16,91,113]. Like many IFN-I-induced proteins, APOL1 and APOL3 may participate in the control of viral replication. Accordingly, APOLs are involved in either pro- or antiviral activities, depending on the virus type [3]. Whereas APOL1 appears to restrict HCV and HIV replication, APOL1 and APOL3 promote the replication of at least four flaviviruses (YFV, WNV, ZIKV and DENV) and two unrelated RNA viruses (VSV and measles) [114,115,116]. In mice, APOL9, a probable APOL1 homologue [6], interacts with PHB2 to restrict Theiler’s virus replication [2]. Given the function of PHB2 as a mitophagy receptor, the antiviral effect of APOL9 probably results from promotion of mitophagy. Accordingly, APOL9 interacts with different proteins and phospholipids involved in autophagy [117,118].

Several viruses require PI4KB activity for mito/autophagosome-derived building of membranous structures, such as replication organelles or inclusion bodies, where new virions can be assembled [3,119]. Thus, PI(4)P is considered as a key building material for the construction of viral replication platforms [119]. This is particularly the case for enteroviruses and various picornaviridae, which recruit PI4KB by hijacking different PI4KB-binding proteins, such as ACBD3 or c10orf76. Other viruses, such as hepatitis C, coronavirus or parainfluenza type 3, also depend on PI4KB for their replication [3].

Mitochondrial fusion, which involves APOL1 and APOL3, facilitates the oligomerization of the Anti-Viral Signaling transmembrane protein MAVS into active complexes with maximum signaling capacity, as well as proper interactions between MAVS and other innate immunity adaptor proteins, such as STING [77,120,121,122]. Moreover, MAVS acts as a potential mitophagy receptor to maintain mitochondrial homeostasis [123]. Thus, antiviral signaling and mitophagy are connected. Accordingly, growing evidence suggests that some viruses promote self-replication through regulating mitophagy-mediated innate immunity [124,125,126,127].

In addition to the control of innate immunity initiation, APOL1 or APOL3 may be required for induction of the T cell response, because in mice, the APOL1- and APOL3-like APOL7C is required for the presentation of phagocytosed antigens at the surface of dendritic cells [92].

In conclusion, interference with APOL1 or APOL3 activities can affect viral infection through their influence on virus replication, but also through their involvement in antiviral signaling and the building of the immune response. Therefore, individuals devoid of APOL1 or APOL3, or expressing the APOL1 variants G1 or G2, may exhibit defects in resistance to viral infection, but also reduced ability to respond to pathogens in general.

## 6. APOLd1 Role in Angiogenesis

APOL1 domain-containing protein d1 (APOLd1), which exhibits some similarities with APOL1, is specifically expressed in endothelial cells, particularly following growth stimulation or ischemic hypoxia [128,129,130,131]. APOLd1 is dispensable for development, but it is strongly induced by inflammation and contributes to promoting autophagic flux. More importantly, APOLd1 is a key regulator of angiogenesis under pathological conditions. Thus, APOLd1 controls endothelial cell signalling and vascular function.

Despite a generally low level of sequence identity, two APOLd1 regions concentrate their homology with APOL1, and these regions precisely correspond to distinct APOL1 domains (Figure 7).

The first domain includes helix 5 and the transmembrane hairpin helix. In APOL1 or APOL3, the defined functions of this region are transmembrane insertion and membrane fusion activity [6,15,16]. Through its high content of positively charged amino acids, helix 5 exhibits a typical site of interaction with anionic lipids such as phosphoinositides. The APOLd1 double-stranded helix hairpin is expected to allow transmembrane insertion at neutral pH, because it is devoid of acidic residues, in contrast to APOL1. A role in membrane fusion is suggested by both the interaction potential of helix 5 and the high flexibility of the two transmembrane spans, owing to a high proportion of β-branched (Ile/Val) and helix-breaking (Gly/Pro/Ser/Thr) amino acids (13/24 and 16/26, for spans 1 and 2, respectively). Such flexibility is essential for membrane fusion [64]. APOLd1 is involved in the organization of endothelial cell–cell junctions, notably through interaction with the cytoskeleton [130], an activity that promotes angiogenesis [131,132]. Thus, it can be hypothesized that the intercellular junction activity of APOLd1 involves membrane fusion.

The function of the second APOL1 domain is less obvious. In APOL1, this domain interacts with the HC1-LZ1 region, except under acidic conditions, in which it becomes accessible and could be induced to play a role in cation-pore-forming activity [7,13,14,16]. In APOLd1, the HC1-LZ1 region is missing, and the residues involved in pH gating of the APOL1 pore are absent (Figure 7). Thus, APOLd1 is unlikely to fold through LZ-driven cis interaction, and it is unlikely to exert pore-forming activity. Moreover, APOLd1 is unlikely to control PI4KB activity, which involves the HC1-LZ1 region in APOL3. In addition, the APOL1 CRAC-2 motif is absent from the APOLd1 sequence (Figure 7), and no other CRAC is predicted in the entire APOLd1, suggesting no cholesterol-binding capacity.

In APOL3, LZ2 interacts with helix 5, apparently preventing protein binding to helix 5 in the absence of helix 5 interaction with VAMP8 for membrane fusion [16]. Therefore, the second APOL1-like domain of APOLd1 could similarly control helix 5 accessibility for fusogenic interaction with VAMP8. Like that which occurs in APOL3, antisense interaction between APOLd1 LZ2 and helix 5 could be driven by pairing between heptad repeats of hydrophobic residues (Figure 7). Such a function would be coherent with the simultaneous presence of both APOL1-like domains in APOLd1.

As APOLd1 shares at least two distinct domains with APOL1, this protein can be considered as an APOL family member, albeit very distant. Whether it represents an ancestor or a derived isoform remains to be determined.

## 7. mAPOL6 Role in Adipogenesis

APOL6, another distant family member devoid of the HC1-LZ1 region but containing a putative fusogenic domain (helix 5 + transmembrane hairpin helix), is specifically expressed in adipocytes. Its activity was particularly analyzed in mice, in which an APOL6 homologue, mAPOL6, is present. This isoform is specifically expressed in adipocytes, and it is involved in adipogenesis control [133] (C. Vermeiren and E. Pays: http://hdl.handle.net/2013/ULB-DIPOT:oai:dipot.ulb.ac.be:2013/279701 (accessed on 7 November 2024).

In mouse lipid droplets, mAPOL6 promotes triglyceride accumulation through the activity of its specific C-terminal region, which triggers the inhibition of lipolysis [133]. The expression of mAPOL6 is induced by IFNγ, and the mAPOL6’ involvement in lipid droplet expansion resulting from high fat diet coincides with the induction of low-grade inflammation of the adipose tissue (C. Vermeiren and E. Pays). Thus, mAPOL6 appears to control inflammation-induced increase in adipocyte size during high fat diet. Interestingly, mAPOL6 is strongly associated with myosin-10 (MYO10), both the heavy chain MYH10 and each of the two light chains [6], suggesting a function linking organelle size control with membrane dynamics. Consistently, MYO10 was found to govern both adipocyte adipogenesis and PI(4)P-mediated lipid droplet dynamics [134,135].

In conclusion, mAPOL6 could interact with MYO10 to control the size of lipid droplets during inflammatory high fat uptake in adipocytes.

## 8. APOL4 Role in Brain Pathology

The expression of APOL1, APOL2 and APOL4 is increased in the brains of schizophrenic patients, and sequence polymorphism in the APOL2-APOL4 intergenic region is associated with a risk for schizophrenia [136,137]. Moreover, APOL4 levels correlate with the tumor progression of gliomas [138]. Thus, increased APOL4 expression correlates with brain pathology.

APOL4 shares high sequence homology with APOL3 (54.2% identity/69.3% similarity), particularly within the three-helical 2–4 bundle that interacts with PI4KB and NCS1 in APOL3 [16] (73.9% identity/88.6% similarity). Moreover, the APOL4 TM hairpin is predicted to insert into membranes at neutral pH as occurs in APOL3, and the APOL4 hairpin is flanked by APOL3-like helices. Therefore, like APOL3, APOL4 may exert interactions with NCS1/PI4KB and VAMP8, potentially inducing membrane fission/fusion activities. In the brain, NCS1 and PI4KB are crucially involved in neurotransmission and synaptic growth [3,139,140], two processes seemingly controlled by APOL8 in mice [141]. In tumors, NCS1 and PI(4)P promote metastasis, likely through the activity of the PI(4)P-binding protein GOLPH3, particularly in gliomas [3,51,53,142,143].

In conclusion, APOL4 could be responsible for APOL3-like membrane remodeling activities in the brain, and inflammation-driven deregulation of APOL4 could cause brain pathology. In terms of clinical implications, APOL4 is considered as a prognostic biomarker for gliomas, and anti-APOL4 therapies could prevent glioma progression.

## 9. Conclusions

The presence of a highly flexible double-stranded hairpin helix with transmembrane potential is the major characteristic of APOLs, including the distantly related APOLd1. This hairpin helix is flanked by SNARE-like α-helices able to interact with the SNARE protein VAMP8, conferring the ability of APOLs to promote membrane fusion, as demonstrated for APOL1 and APOL3. However, in the case of APOL1, the fusion-promoting activity is pH-dependent, and absolutely requires acidic conditions. Moreover, whereas both APOLs can bind to anionic lipids such as phosphoinositides and cardiolipin, only APOL1 contains cholesterol-binding motifs, one of which is only accessible under acidic conditions due to pH-dependent structural folding. Thus, APOL1 appears to have been conceived for functions requiring acidic pH. Accordingly, APOL1 can kill African trypanosomes in a process involving cholesterol-associated APOL1 trafficking in parasite acidic endosomes. In human podocytes, only APOL3 exhibits membrane fusion activity, which allows mitophagy completion.

In addition to its membrane fusion activity, APOL3 also exhibits specific interactions with PI4KB- and PI4KB-controlling proteins. These interactions allow PI(4)P synthesis, which is involved in vesicular secretion at the Golgi, mitochondrial fission, mitophagy, apoptosis, and phagolysosome formation, possibly linked to antigen cross-presentation in dendritic cells. The lack of APOL1 interaction with PI4KB and, hence, the lack of APOL1 activity in membrane fission, results from APOL1-specific pH-dependent folding, which prevents the accessibility of hydrophobic N-terminal helices. Such folding may conversely allow APOL1 to play the distinct role of driving APOL3 and PI4KB trafficking in ATG9A vesicles from the Golgi to MERCSs, through interactions with the NM2A myosin and mitophagy receptor PHB2. This traffic is triggered by INF-I signaling resulting from pathogen infection, and induces mitophagy and apoptosis to cope with mitochondrial damage due to inflammation. Thus, APOL1 and APOL3 are involved in distinct activities controlling membrane dynamics in response to pathogen infection and, consequently, innate immunity induction. Such activities appear to be particularly important in kidney biology, presumably because podocyte filtration activity crucially involves PI(4)P- and cholesterol-dependent actomyosin control of membrane dynamics. This probably explains why the prominent pathology resulting from interference with APOL1 and APOL3 functions, as occurs following the expression of the APOL1 C-terminal variants G1 or G2, is nephropathy.

## 10. Future Research Directions and Key Outstanding Questions

The most obvious issue in the field is the definition of the mechanism by which the APOL1 variants G1 and G2 induce podocyte cytotoxicity. Either direct evidence for APOL1 insertion as a cation pore in the podocyte plasma membrane should be provided, or the role of G1 or G2 in activating the TRPC6 and BK cation channels should be demonstrated. The hypothetical interaction of WT, G1 and G2 APOL1 with cholesterol should be analyzed and compared. Moreover, the involvement of each CRAC motif in podocyte toxicity needs to be assessed.

Related to the previous issue, the exact structure and function of the APOL1 cation pore remain to be established. Is transmembrane insertion of the HC2 region required for pH gating of the pore, or does this region orient cations to the pore when present in the pore vestibule? Is the APOL1 pore-forming activity uniquely operating in trypanosome acidic compartments, or is it also playing a non-recognized role in the podocyte cytoplasm? Would this activity represent a non-physiological effect of in vitro conditions, like what appears to be the case of a similar activity in pro- and anti-apoptotic proteins of the BCL2 (B-cell Lymphoma 2) family, which contain an APOL-like transmembrane helix hairpin?

Among the most striking features of APOLs, at least for APOL1 and APOL6, interaction with non-muscular myosins (NM2A, MYO10) is worth noticing. How APOLs could be involved in myosin functions (traffic and membrane fission) is totally unknown. APOL1’s interaction with the NM2A Regulatory Light Chain (RLC) suggests the involvement of APOLs in myosin activity regulation. Such interaction is consistent with the similarity of the myosin RLC with calmodulin-like proteins such as NCS1, which can strongly bind to APOL3. Interestingly, poly(I:C) treatment triggers the selective dissociation of phosphorylated RLC from APOL1, implying that RLC-APOL1 interaction could be modulated by inflammation (E. Pays, unpublished). Moreover, the interaction of NM2A with PI(4)P triggers the release of RLC from the myosin complex [48], suggesting that APOL3 could interfere with myosin activity through its involvement in PI(4)P synthesis. The definition of the APOL1 binding site in the NM2A RLC, possibly at the RLC interface with the myosin heavy chain, as well as the effects of PI(4)P and inflammation on this interaction, clearly deserves investigation.

Along the same line, the effect of inflammation-driven ARF1 activation on the dissociation of the APOL3/PI4KB and APOL1/NM2A complexes from the Golgi is worth considering. The hypothesis that activated ARF1 competes with APOL3 for PI4KB binding should be tested experimentally.

Another interesting issue is the possible role of APOLs in the control of gene expression. PI4KB, APOL1 and APOL3 contain nuclear localization signals and are present in both the nucleus and the cytoplasm [15,16,144,145]. Moreover, nuclear PI4KB and PI4P are associated with mRNA speckles and factors involved in pre-mRNA splicing or transport [146,147], and both APOL1 and mAPOL9 immunoprecipitates contain RNA processing components [2,15]. Thus, APOL1, APOL3 and PI4KB may influence gene expression through the control of mRNA editing and mRNA delivery to the cytoplasm. Consistently, NCS1 deficiency was reported to affect the mRNA levels of genes involved in mitochondrial activity [148], and the transcriptome of G1 or G2 podocytes exhibited specific differences from that of WT cells [149]. Given the existence of natural APOL1 KO and APOL3 KO individuals [28,29,40], transcriptome analysis of podocytes from these individuals is worth performing.

Finally, the mechanism controlling the cellular distribution of each APOL isoform remains to be identified. All isoforms may share similar membrane-remodeling activities but participate in different organ-specific processes, depending on the isoform organ-specific expression.

## Figures and Tables

**Figure 1 cells-13-02115-f001:**
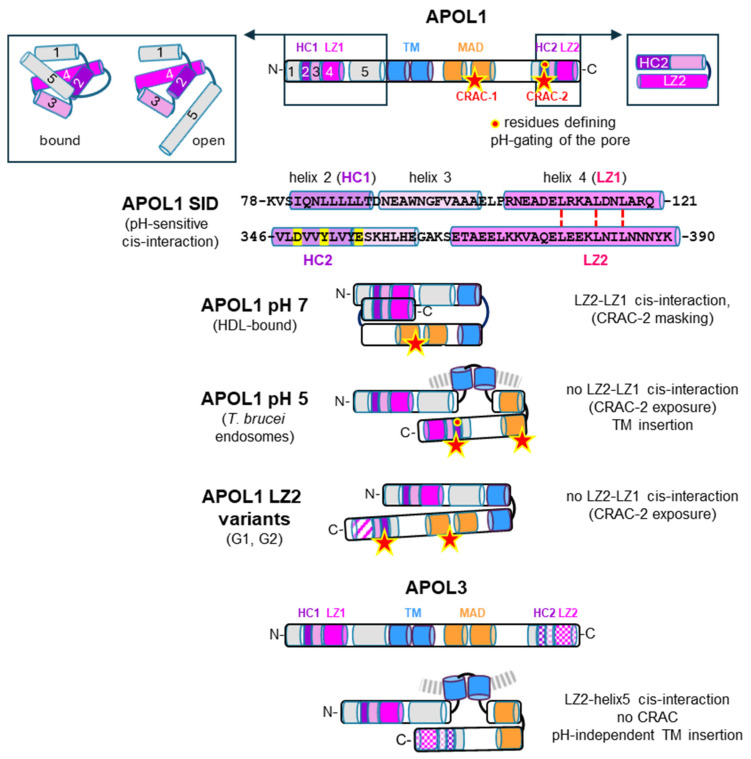
Structural features of APOL1 and APOL3. The colored cylinders represent different α-helices, some of which are numbered, according to Ultsch et al. [5]. HC1, HC2 = hydrophobic clusters 1, 2; LZ1, LZ2 = leucine zippers 1, 2; CRAC-1, CRAC-2 = cholesterol recognition amino acid consensuses 1, 2 (represented by red stars); TM = potential transmembrane hairpin helix; MAD = membrane-addressing domain. At acidic pH, the APOL1 TM hairpin can form weak anion pores, but pH neutralization confers high cation conductance. HC2 amino acids involved in pore pH-gating are highlighted in yellow. The boxes illustrate the folding of the N- and C-terminal APOL1 domains. In the isolated N-terminal domain, helix 5 can adopt two positions, preventing (bound) or not preventing (open) helix 4 accessibility [5]. APOL1 SID represents the Smallest Interacting Domain between N- and C-terminal regions. This interaction, driven by LZ1-LZ2 pairing, is affected either by acidic conditions, as in trypanosome endosomes, or by LZ2 mutations, as in the natural G1 or G2 variants. In APOL3, LZ2 interacts with helix 5.

**Figure 2 cells-13-02115-f002:**
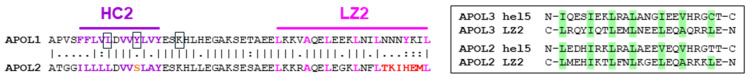
APOL2 sequence comparison with APOL1 and APOL3. Hydrophobic residues characterizing HC2 and LZ2 are highlighted in violet and pink, respectively. APOL1 CRAC-2 residues are boxed. Key APOL2 HC2 and LZ2 differences from APOL1 are in orange and red, respectively. The boxed sequence alignments show antisense pairing between helix 5 and LZ2, based on hydrophobic heptad repeats (highlighted in green).

**Figure 3 cells-13-02115-f003:**
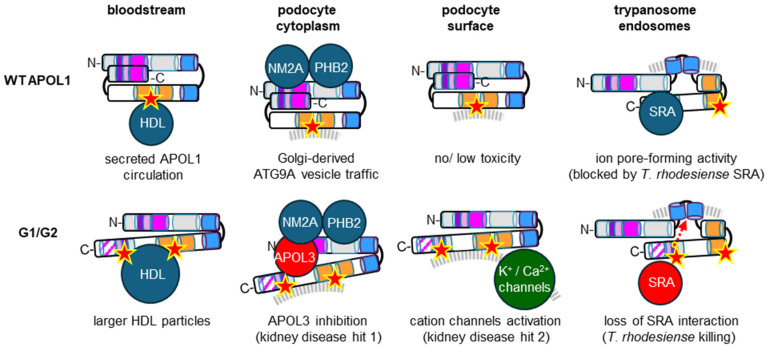
WT or C-terminal variant APOL1 interactions and activities. The same symbols and colors as in Figure 1. In the last scheme, hypothetical cation driving to the membrane pore at neutral pH [14] is symbolized by a dotted red arrow.

**Figure 4 cells-13-02115-f004:**
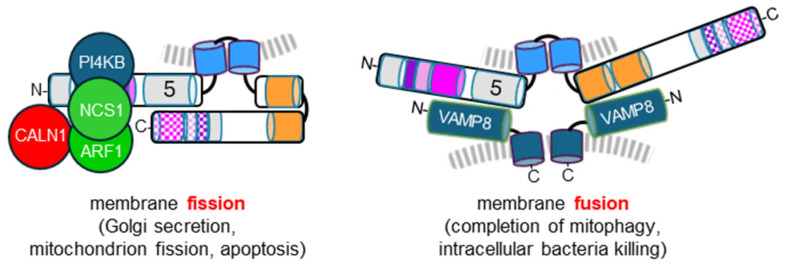
APOL3 interactions and activities. The same symbols, colors and numbers as in Figure 1. NCS1 and CALN1 are alternative APOL3 binders activating or inhibiting PI4KB, depending on calcium concentration. ARF1 binds to APOL3, and inflammation-mediated ARF1 activation promotes its binding to PI4KB, possibly dissociating APOL3-PI4KB interaction. VAMP8 interacts with both helices 4–5 and MAD, promoting membrane fusion.

**Figure 5 cells-13-02115-f005:**
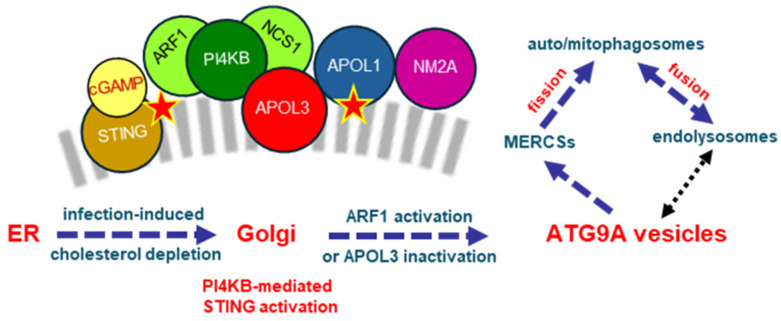
Intracellular traffic of proteins involved in infection-induced changes in membrane dynamics. Detection of pathogen DNA triggers the synthesis of cyclic GMP-AMP (cGAMP), which binds to STING and disrupts STING-cholesterol interactions, allowing STING binding to PI(4)P for translocation to the Golgi. In the Golgi, STING undergoes oligomerization, which induces IFN-I inflammatory signaling. IFN-I activates ARF1, leading to STING, PI4KB and APOL3 dissociation from the Golgi in ATG9A vesicles trafficking to MERCSs, promoting membrane fission and fusion events linked to auto/mitophagy and apoptosis. This pathway allows inflammation termination due to STING autophagic degradation. Through association with NM2A and PHB2, APOL1 could direct ATG9A vesicles to MERCSs, where mitophagy is initiated. Red stars represent cholesterol interactions. The double-arrowed black dotted line represents the involvement of endolysosomes in both mitochondrion fission and autophagosome formation by ATG9A vesicles.

**Figure 6 cells-13-02115-f006:**
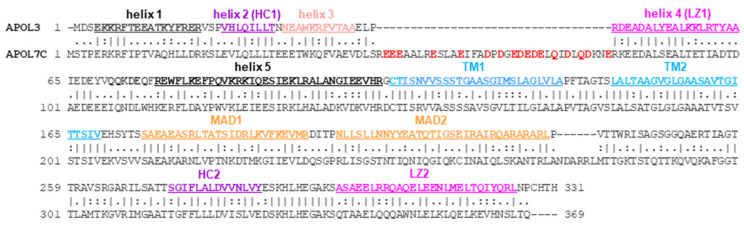
Sequence alignment between human APOL3 (above) and mouse APOL7c (below), using Clustal Omega (https://www.ebi.ac.uk/Tools/msa/clustalo/ (accessed on 4 November 2024)). Insertion of clustered acidic residues, highlighted in red, characterizes the murine APOL7 family.

**Figure 7 cells-13-02115-f007:**
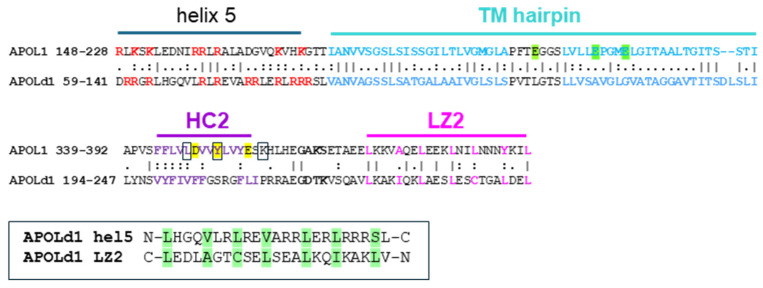
The two APOL1-like domains of APOLd1. Positively charged residues of helix 5 are highlighted in red, and the two helices of the putative transmembrane domain are highlighted in blue, with acidic residues in green. Hydrophobic residues characterizing the HC2 and LZ2 helices are highlighted in violet and pink, respectively. The amino acids involved in pH gating of the APOL1 pore are highlighted in yellow. The APOL1 residues defining CRAC-2 are boxed, and the loop sequences between the two helices of the double-stranded HC2-LZ2 helix hairpin are in bold. The boxed sequence alignment shows antisense pairing between APOLd1 helix 5 and LZ2, based on hydrophobic heptad repeats (highlighted in green).

## Data Availability

No new data were created or analyzed in this study. Data sharing is not applicable to this article.

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
