# Peer review of "Apolipoprotein-L Functions in Membrane Remodeling"

_cells, 2024, doi:10.3390/cells13242115_

Round 1
Reviewer 1 Report
Comments and Suggestions for Authors
This review discusses the role of APOL in inflammatory responses from a number of different perspectives and is very interesting.
However, it also raises a number of issues.
The text states that APOL is a hydrophobic protein, but like other apolipoproteins it has an α-helix structure and is probably an amphiphilic protein with both hydrophilic and hydrophobic properties.
The author should include some more basic information about APOL.
The details of APOL for expression in organs and cells, its presence in the blood, and its presence and relevance at sites of inflammation should be described in more detail.
The article mentions that APOL is present in the blood by binding to lipoproteins, especially HDL. It would be better to describe the localization in the blood in more detail. For example, localization to other lipoproteins or within lipoproteins, LDL and other lipoproteins other than HDL. Also, if there is any knowledge about the localization of APOL in HDL, such as what type of HDL particles (e.g. small or large particles) it is localized in, or how it changes during inflammation, it would be good to include this information.
Author Response
Cells-3339577. Reply to reviewers
General comment: precision on the focus of the review.
As an introductory paper in the Cells Special Issue “Evolution, Structure and Functions of APOLs”, the aim of this review is to update information about the basic function of (all) APOLs, as rightly summarized by Reviewer 1 (The paper reviews the structure, molecular interactions, and functions of APOL isoforms in membrane dynamics associated with inflammation. It further focuses on their roles in pathogen infection, angiogenesis, and adipogenesis). This precision is important in view of some reviewer’s comments.
Reviewer 1
This review discusses the role of APOL in inflammatory responses from a number of different perspectives and is very interesting.
However, it also raises a number of issues.
The text states that APOL is a hydrophobic protein, but like other apolipoproteins it has an α-helix structure and is probably an amphiphilic protein with both hydrophilic and hydrophobic properties.
The author should include some more basic information about APOL.
The details of APOL for expression in organs and cells, its presence in the blood, and its presence and relevance at sites of inflammation should be described in more detail.
The article mentions that APOL is present in the blood by binding to lipoproteins, especially HDL. It would be better to describe the localization in the blood in more detail. For example, localization to other lipoproteins or within lipoproteins, LDL and other lipoproteins other than HDL. Also, if there is any knowledge about the localization of APOL in HDL, such as what type of HDL particles (e.g. small or large particles) it is localized in, or how it changes during inflammation, it would be good to include this information.
Reply
Regarding hydrophobicity, APOLs must globally be hydrophobic, because they clearly interact with lipids as demonstrated for APOL1 (firstly identified as cholesterol-associated protein; hydrophobicity measured in ref. 15) and APOL3 (exposure of hydrophobic clusters for binding to hydrophobic regions of PI4KB and NCS1). These characteristics, as well as the structure of the different helices, are mentioned in detail in the text and figures.
Request for more basic information and details on expression and presence at sites of inflammation: I sincerely think that I presented all known information on these proteins. I just completed this information by mentioning the ubiquitous expression of APOLs in all organs. This reviewer should be aware that this field is recent, poorly documented and only emerging, following the discovery that APOL1 variants cause kidney disease (hence the interest of this review).
Binding to lipoproteins: I do not see what needs to be added: APOL1 was identified as specifically associated with HDL-C particles (as mentioned in the text): HDL-C is the densest fraction of HDLs. Regarding APOL1 changes in HDL-C particles during inflammation, this information in now given in the new section 4, added following the requests by reviewer 2 (inflammation is linked to reduction of HDL-C levels).
Reviewer 2 Report
Comments and Suggestions for Authors
This review of APOLs is well-developed and written. However, I notice the absence of a detailed discussion about the role of APOLs in kidney diseases, in contrast to the focus on their role in pathogen infection. Addressing this would be particularly important, as the conclusions mention nephropathy and several references highlight their involvement in podocyte function.
In my opinion, the review is well-written and informative. Below, I provide more details and address your questions:
• What is the main question addressed by the research? The paper reviews the structure, molecular interactions, and functions of APOL isoforms in membrane dynamics associated with inflammation. It further focuses on their roles in pathogen infection, angiogenesis, and adipogenesis.
• Do you consider the topic original or relevant to the field? Does it address a specific gap in the field? Please also explain why this is/is not the case.
In my opinion, the review is relevant but not entirely original, as the author has previously published other reviews on APOLs, for example, in FEBS J. 2020 ("The function of apolipoproteins L (APOLs): relevance for kidney disease, neurotransmission disorders, cancer, and viral infection") and in Cells 2024 ("Apolipoprotein-L1 (APOL1): From Sleeping Sickness to Kidney Disease"). Other authors have also reviewed this topic (PMID: 38415700, PMID: 38341125, PMID: 37261508, etc.). Nevertheless, I enjoyed reading this review.
• What does it add to the subject area compared with other published material?
I particularly appreciated the detailed molecular descriptions and insights into APOL interactions. As this is a review, its primary purpose is to provide a comprehensive update on the current knowledge rather than introduce novel findings, which it achieves well.
• Are the conclusions consistent with the evidence and arguments presented, and do they address the main question posed? Please also explain why this is/is not the case.
In my opinion, the conclusions are generally consistent with the evidence and arguments presented. However, I found the lack of focus on their role in kidney diseases and potential therapies to be a notable omission.
• Are the references appropriate?
Yes, the references are appropriate.
• Any additional comments on the tables and figures: The figures are descriptive and well-presented. However, there are no tables included in the review.
Author Response
Cells-3339577. Reply to reviewers
General comment: precision on the focus of the review.
As an introductory paper in the Cells Special Issue “Evolution, Structure and Functions of APOLs”, the aim of this review is to update information about the basic function of (all) APOLs, as rightly summarized by Reviewer 1 (The paper reviews the structure, molecular interactions, and functions of APOL isoforms in membrane dynamics associated with inflammation. It further focuses on their roles in pathogen infection, angiogenesis, and adipogenesis). This precision is important in view of some reviewer’s comments.
Reviewer 2
This review of APOLs is well-developed and written. However, I notice the absence of a detailed discussion about the role of APOLs in kidney diseases, in contrast to the focus on their role in pathogen infection. Addressing this would be particularly important, as the conclusions mention nephropathy and several references highlight their involvement in podocyte function.
In my opinion, the review is well-written and informative. Below, I provide more details and address your questions:
- What is the main question addressed by the research? The paper reviews the structure, molecular interactions, and functions of APOL isoforms in membrane dynamics associated with inflammation. It further focuses on their roles in pathogen infection, angiogenesis, and adipogenesis.
- Do you consider the topic original or relevant to the field? Does it address a specific gap in the field? Please also explain why this is/is not the case.
In my opinion, the review is relevant but not entirely original, as the author has previously published other reviews on APOLs, for example, in FEBS J. 2020 ("The function of apolipoproteins L (APOLs): relevance for kidney disease, neurotransmission disorders, cancer, and viral infection") and in Cells 2024 ("Apolipoprotein-L1 (APOL1): From Sleeping Sickness to Kidney Disease"). Other authors have also reviewed this topic (PMID: 38415700, PMID: 38341125, PMID: 37261508, etc.). Nevertheless, I enjoyed reading this review.
- What does it add to the subject area compared with other published material?
I particularly appreciated the detailed molecular descriptions and insights into APOL interactions. As this is a review, its primary purpose is to provide a comprehensive update on the current knowledge rather than introduce novel findings, which it achieves well.
- Are the conclusions consistent with the evidence and arguments presented, and do they address the main question posed? Please also explain why this is/is not the case.
In my opinion, the conclusions are generally consistent with the evidence and arguments presented. However, I found the lack of focus on their role in kidney diseases and potential therapies to be a notable omission.
- Are the references appropriate?
Yes, the references are appropriate.
- Any additional comments on the tables and figures: The figures are descriptive and well-presented. However, there are no tables included in the review.
Reply
In a previous paper, duly referenced here (ref. 12), I have extensively discussed the role of APOLs in kidney disease. However, given this request, I now added a specific paragraph summarizing this debate, together with clinical implications (new Section 4: APOL1 and APOL3 roles in kidney disease).
Regarding the originality of the current review, it is double: (1) updating the current information with very recent references, particularly regarding APOL1, APOL3 and APOL7C, and (2) presenting comprehensive information on the structure and role of all members of the APOL family. In this regard, I added now a section on the APOL4 role in brain pathology
Reviewer 3 Report
Comments and Suggestions for Authors
Key weakness is lack of critical evaluation of methodological approaches used in cited studies. While paper effectively reports findings, it would benefit from dedicated section assessing quality of evidence, discussing limitations of experimental approaches, and identifying potential biases in research.
Significant limitation is paper's tendency to present findings sequentially without deeply analyzing contradictions between studies. Review would be more valuable if it explicitly discussed contradictory results, analyzed possible reasons for discrepancies, and suggested ways to resolve conflicting evidence. This approach would provide readers with more nuanced understanding of current state of knowledge in field.
Review focuses heavily on molecular mechanisms while providing insufficient discussion of clinical implications. Dedicated section addressing how basic science findings translate to clinical applications and therapeutic possibilities would enhance relevance to medical practice. Additionally, paper sometimes presents single explanations for phenomena without adequately considering alternative mechanisms. More systematic consideration of alternative hypotheses and competing theories would provide more balanced perspective.
Figures present another area for improvement. Some are complex and difficult to interpret without extensive reading of text. Simplifying figures, adding more explanatory legends, and including summary diagrams that integrate key concepts would make content more accessible. Review would also benefit from more thorough discussion of technical challenges and limitations in studying APOL proteins, along with potential solutions to these challenges.
Notable omission is limited discussion of future research directions and key outstanding questions. Adding comprehensive section on future research priorities would help guide subsequent investigations in field. Review could also better integrate APOL biology with broader cellular and physiological processes, providing context for how APOL functions relate to other cellular pathways and disease mechanisms.
Author Response
Cells-3339577. Reply to reviewers
General comment: precision on the focus of the review.
As an introductory paper in the Cells Special Issue “Evolution, Structure and Functions of APOLs”, the aim of this review is to update information about the basic function of (all) APOLs, as rightly summarized by Reviewer 1 (The paper reviews the structure, molecular interactions, and functions of APOL isoforms in membrane dynamics associated with inflammation. It further focuses on their roles in pathogen infection, angiogenesis, and adipogenesis). This precision is important in view of some reviewer’s comments.
Reviewer 2
Key weakness is lack of critical evaluation of methodological approaches used in cited studies. While paper effectively reports findings, it would benefit from dedicated section assessing quality of evidence, discussing limitations of experimental approaches, and identifying potential biases in research.
Significant limitation is paper's tendency to present findings sequentially without deeply analyzing contradictions between studies. Review would be more valuable if it explicitly discussed contradictory results, analyzed possible reasons for discrepancies, and suggested ways to resolve conflicting evidence. This approach would provide readers with more nuanced understanding of current state of knowledge in field.
Review focuses heavily on molecular mechanisms while providing insufficient discussion of clinical implications. Dedicated section addressing how basic science findings translate to clinical applications and therapeutic possibilities would enhance relevance to medical practice. Additionally, paper sometimes presents single explanations for phenomena without adequately considering alternative mechanisms. More systematic consideration of alternative hypotheses and competing theories would provide more balanced perspective.
Figures present another area for improvement. Some are complex and difficult to interpret without extensive reading of text. Simplifying figures, adding more explanatory legends, and including summary diagrams that integrate key concepts would make content more accessible. Review would also benefit from more thorough discussion of technical challenges and limitations in studying APOL proteins, along with potential solutions to these challenges.
Notable omission is limited discussion of future research directions and key outstanding questions. Adding comprehensive section on future research priorities would help guide subsequent investigations in field. Review could also better integrate APOL biology with broader cellular and physiological processes, providing context for how APOL functions relate to other cellular pathways and disease mechanisms.
Reply
As far as I could see I did not find contradictions in the field, except for the notable and important debate on the APOL1 role in kidney disease, where different interpretations were given to the same experimental evidence (cation pore-forming activity induced by APOL1 variants). I have extensively discussed these contradictory interpretations in a previous paper, duly referenced here (ref. 12). However, given this comment and the request by Reviewer 1, I now added a specific section (4) summarizing this debate, together with clinical implications. Similarly, I now mentioned the clinical implications of APOL4 activity, in a newly added section on this isoform (section 8).
Regarding methodological issues, I did not identify any question to debate. Methodological issues should have been discussed during reviewing of each original paper, and it would be very tedious to me to search for eventual problems in the 147 cited references. To be useful, this request should be accompanied by mentioning where exactly methodological approaches should be discussed.
Along the same line, speculating alternative interpretations in the general absence of contradicting views appears to be useless, and would unnecessarily load the text without precise reason. Apart from the addition of section 4 now discussing the main controversy in the field, I would like to point out that in several sections, I speculated in length on the possible activities of the different APOL isoforms, exclusively based on experimental evidence. In my opinion, adding further speculation for sake of imagining possible alternatives, would complicate and weaken the message.
Regarding future research priorities and APOL biology integration with broader cellular and physiological processes, I added a comprehensive section (section 10) at the end, with a final paragraph summarizing the global function of APOLs in different tissues.
Regarding the Figures, I do not see easy ways to satisfy this reviewer. I am aware of the complexity of the subject, and I did my best to be as clear as possible. Other reviewers did not raise this point.
Reviewer 4 Report
Comments and Suggestions for Authors
The manuscript claims to be an extensive review on the role of apolipoproteins L in immunological processes via cellular membranes remodelling. In its current form the text is hard to follow and there is a content imbalance between the description of the structural/genetic features of ApoLs and their per se function in the inflammatory processes. Being an informative review with a focused topic on the inflammatory mechanisms modulated by apolipoproteins I would suggest a significant shrinking of the overall text and especially the removal of many parts pertaining to genetic sequences and structural details with derail the attention from the main purpose of the review: to offer a valuable update of the immunological role of ApoLs. I would also add more text about the general mechanisms of inflammation and offer at least a background description of the antigen presentation processes in the context of the MHC molecules in different cell types (not only DCs)[if that kind of information is available] and insist on the mechanistic aspects of ApoLs pertaining to these immune processes.
In conclusion, the manuscript is hard to follow and the discussion derails from the main focus expressed in the title. The review would significantly benefit if some parts of the text (related to the structure and genetics of ApoLs) would be compressed/cut/re-written and some other parts would be expanded, especially those related to the immune mechanisms of the inflammatory response.
Author Response
Cells-3339577. Reply to reviewers
General comment: precision on the focus of the review.
As an introductory paper in the Cells Special Issue “Evolution, Structure and Functions of APOLs”, the aim of this review is to update information about the basic function of (all) APOLs, as rightly summarized by Reviewer 1 (The paper reviews the structure, molecular interactions, and functions of APOL isoforms in membrane dynamics associated with inflammation. It further focuses on their roles in pathogen infection, angiogenesis, and adipogenesis). This precision is important in view of some reviewer’s comments.
Reviewer 3
The manuscript claims to be an extensive review on the role of apolipoproteins L in immunological processes via cellular membranes remodelling. In its current form the text is hard to follow and there is a content imbalance between the description of the structural/genetic features of ApoLs and their per se function in the inflammatory processes. Being an informative review with a focused topic on the inflammatory mechanisms modulated by apolipoproteins I would suggest a significant shrinking of the overall text and especially the removal of many parts pertaining to genetic sequences and structural details with derail the attention from the main purpose of the review: to offer a valuable update of the immunological role of ApoLs. I would also add more text about the general mechanisms of inflammation and offer at least a background description of the antigen presentation processes in the context of the MHC molecules in different cell types (not only DCs)[if that kind of information is available] and insist on the mechanistic aspects of ApoLs pertaining to these immune processes.
In conclusion, the manuscript is hard to follow and the discussion derails from the main focus expressed in the title. The review would significantly benefit if some parts of the text (related to the structure and genetics of ApoLs) would be compressed/cut/re-written and some other parts would be expanded, especially those related to the immune mechanisms of the inflammatory response.
Reply
In my opinion, these comments reflect a misunderstanding of the message of this review, because this reviewer is not interested in APOLs per se, but in immunological mechanisms where APOLs could be involved. The main purpose of the review is not “to offer a valuable update of the immunological role of ApoLs”, but to offer an update on the activity of APOLs, which I am proposing to consist in membrane remodelling (not discussed before), already under normal conditions (secretion), but more particularly in the process of inflammation. Thus, the focus is not on the role of APOLs in inflammation, but on the role of APOLs in membrane remodelling, particularly that resulting from inflammation. APOLs expression is increased by inflammation, and through their activity on membrane remodelling, APOLs affect the inflammatory process, as described here for STING and MAVS activation, as well as for antigen cross-presentation. Extending the review on the mechanisms of immunology is clearly outside its scope, and reviewing antigen presentation in other antigen-presenting cells than dendritic cross-presenting CD8a+ cells is irrelevant. In this regard, it is worth stressing that the effect of APOL7C on antigen presentation is not only quite recent (November 1st, 2024), but it is also the only reference linking APOLs with immunological processes apart from their involvement in membrane remodelling for inflammation activation and mitophagy induction. Thus, in the absence of additional information, expanding the immunology aspects of APOLs biology appears to be premature and impossible to do. In contrast, cutting parts pertaining to genetic sequences and structural details of APOLs would obviously remove essential information regarding the function of the different APOLs, which is the core of the review. Accordingly, this section was estimated to be the most important by reviewer 1 (“I particularly appreciated the detailed molecular descriptions and insights into APOL interactions.”).
To better state the content of the review, - and avoid any misunderstanding -, I decided to withdraw the mention to inflammation in the title (now: “Apolipoproteins-L functions in membrane remodelling”).
Round 2
Reviewer 1 Report
Comments and Suggestions for Authors
The author has responded appropriately to the reviewer's comments and made revisions.
There are no further revisions or comments.
Reviewer 2 Report
Comments and Suggestions for Authors
The author has adequately addressed the concerns, and the paper is now ready for publication.
Reviewer 4 Report
Comments and Suggestions for Authors
By removing the word 'inflammation' from the title, the entire focus of the review has been changed. The inflammatory mechanisms involve immunological reactions and interactions and this is why in my previous review I suggested a more balanced description between the molecular/structural/genetic aspects pertaining to ApoL, and it's role in immunity, the more so as most of the pathologies described in the review also have an immunological/inflammatory component. However, my current understanding is that the link between ApoL and membrane remodeling in the context of immunological processes is not well characterized and this is why I mentioned in my previous review of the manuscript that such information should be added only 'if available'.
As the current revision of the manuscript offers a new focus and perspective, it would be suitable for publication in the current form.